# Chemical Characterization, Sensory Definition and Prediction Model of the Cider Dryness from New York State Apples

**DOI:** 10.3390/foods12112191

**Published:** 2023-05-30

**Authors:** Monica Picchi, Paola Domizio, Matt Wilson, Josè Santos, Frederick Orrin, Bruno Zanoni, Valentina Canuti

**Affiliations:** 1DAGRI—Department of Agriculture, Food, Environment and Forestry, University of Florence, Via Donizetti, 6, 50144 Firenze, Italy; monica.picchi@unifi.it (M.P.); paola.domizio@unifi.it (P.D.);; 2Enartis, 7795 Bell Road, Windsor, CA 95492, USA; m.wilson.cider@gmail.com (M.W.);; 3Vinquiry Laboratories, 7795 Bell Road, Windsor, CA 95492, USA

**Keywords:** apple cider, dryness, sensory perception, chemical characterization, PLS model, multivariate analysis

## Abstract

Cider is a fermented drink obtained from apple juice. As a function of the used apple cultivar, cider can be classified in four different categories (dry, semi-dry, semi-sweet, sweet), distinguished by the attribute of “dryness,” which reflects the sweetness and softness perceived. The dryness level is defined by scales (IRF, NYCA scales) based on the residual sugar, titratable acidity and tannin contents. Despite some adjustments, these scales show limitations in the prediction of actual perceived dryness, as they cannot consider the complicated interrelation between combined chemical compounds and sensory perception. After defining the perceived sensory dryness and its sensory description by using the quantitative descriptive analysis (QDA) method, a multivariate approach (PLS) was applied to define a predictive model for the dryness and to identify the chemical compounds with which it was correlated. Three models were developed, based on three different sets of chemical parameters, to provide a method that is easily applicable in the ordinary production process of cider. The comparison between the predicted rating and the relative scales scores showed that the models were able to predict the dryness rating in a more effective way. The multivariate approach was found to be the most suitable to study the relation between chemical and sensory data.

## 1. Introduction

Cider, or “hard” cider, is a fermented drink obtained from apple juice. Any apple cultivar can be used, but those that can contribute to the sensory characteristics of ciders are of particular interest [1,2]. Apple cultivars are classified according to their malic acid and tannin concentrations as “sweet” (low in acid, low in tannin), “sharp” (high in acid, low in tannin), “bittersweet” (low in acid, high in tannin), and “bitter-sharp” (high in acid, high in tannin). The high-tannin “bittersweet” and “bitter-sharp” apples are commonly referred to as cider apples and they contribute to the structure, preservation, finish and perceived complexity of the cider. In the United States, the availability of cider apples is limited, so many cider producers use a combination of low-tannin fruit and high-tannin cider-apple-juice concentrate or other tannin-rich adjuncts to meet the growing demand for cider in the marketplace [3].

According to the most recent survey, conducted in May 2023, there were 1.071 cider producers based in the United States. Most of these cider-producing companies were traditional cideries. The largest collection of U.S. cider-producing companies can be found in New York state, which accounted for 106 of the total (https://www.ciderguide.com accessed on 21 May 2023).

Cider quality and its sensory characteristics are related to several important groups of chemical compounds, including polyphenols, sugars and acids.

Concerning phenolic compounds, researchers have found that apples, when compared to 10 other commonly consumed fruits (avocado, banana, blueberry, white grape, grapefruit, lemon, melon, nectarine, orange and peach), had the highest content of soluble free phenols [4]. Each apple cultivar has its own polyphenolic profile, which is also dependent on harvest year, climatic variables, cultivation and storage conditions [5,6]. The polyphenolic composition of ciders mainly depends on the mixture of apple varieties and the cider-making procedures used for their elaboration. Apple polyphenols can be classified into five main groups: phenolic acids, flavanols, anthocyanins, flavonols, dihydrochalcones. Chlorogenic acid is the most abundant of the phenolic acids reported in apples [7]. The flavanols consist of the monomers (+)-catechin and (−)-epicatechin, and procyanidins (also referred to as condensed tannins), which are the oligomers of monomers [8,9,10]. Anthocyanins are another group of polyphenols responsible for skin pigmentation in red cultivars [11]. Flavanols (mainly quercetin derivatives) and dihydrochalcones (mostly phloretin derivatives) are relatively minor contributors to pigmentation and are present in relatively much lower concentrations than flavanols and phenolic acids [12,13,14].

The phenolic content and profile have important effects on the sensory properties of apple ciders, particularly their color, bitterness, and astringency [15]. Given the composition of cider (water, sugar—mainly fructose—organic acid and phenolic compounds), bitterness and astringency have been considered important attributes to define cider quality. These attributes were studied by Lea and Arnold [16] in bittersweet English cider, which demonstrated that no individual procyanidin can be uniquely identified with bitterness or astringency, while bitterness can be associated with oligomeric procyanidins, overall epicatechin tetramer and astringency to higher-molecular-weight procyanidins. Simoneaux et al. [17] studied the taste and tactile characteristics (bitterness, astringency, sweetness and sourness) of cider and highlighted that they were modified according to the concentration of procyanidins. Their study was performed on model solutions with procyanidins extracted from apples at different levels of polymerization and evaluated by a panel of trained judges. The degree of polymerization of the procyanidins influenced only the bitterness and astringency, but this impact was not the same for all concentrations. The same authors, in another study [18], showed that procyanidins exerted a significant negative effect on bitterness and a positive effect on astringency with interactions due to fructose. Moreover, the presence of ethanol significantly increased the perception of bitterness, while the presence of malic acid and fructose decreased it.

Pando Bedriñana et al. [19] studied the perceived characteristics of ice cider by using the citation-frequencies method and an assessment of quality by a group of experts using a five-point discontinuous scale. The results of their study demonstrated that the perceptions of sweetness, acidity and bitterness were closely interrelated and that these, in turn, were related to the contents of sugars, acids, polyphenols and alcoholic degree. According to the study by Riekstina et al. [20], polyphenols showed moderate negative correlation with the perception of fruity aroma. To improve the structure and complexity of cider, Martin et al. [21] enriched a base cider with the same amounts of endogenous and exogenous tannins of three different types (high-tannin cider apple, grape tannin and gallnut). A panel of 193 consumers were recruited to evaluate the quality of the ciders, but no significant differences in overall enjoyment were detected, demonstrating a wide range of astringency acceptability.

To allow consumers to choose according to their preferences, ciders are classified by a marketable scale derived from the traditional sparkling-wine-style sweetness scale (UE reg. n. 607/2009–encl. XIV), based solely on the quantity of residual sugar content and expressed as “dryness.” The limit of this dryness scale was its low correlation with consumer perception, as it does not consider the relation between sensory attributes such as astringency, bitterness and acidity and other chemical compounds, such as tannins and malic acid. Accordingly, two other scales are now available: the International Riesling Scale (IRF) and the New York Cider Association (NYCA) scale. These better relate the perceived dryness to the chemical composition, taking into consideration not only the residual sugar content, but also the titratable acidity, measured as malic acid, and the tannin concentration.

The study of the relationship between chemical composition and sensory perception is complicated by the fact that every sensory attribute is influenced by many variables. To explore this relationship, a multivariate method, such as partial least squares (PLS), can be appropriate. The use of PLS analysis is a “soft modeling” method to extract “factors” or latent variables, which are linear combinations of one set of variables (such instrumental data) able to predict much of the variation in another set of variables (such as sensory-data-attribute ratings) [22]. The PLS model indicates how well the variables in one data set predict the variation among variables in a second set by validation tests, which determine the percentage of variation in one data set that is accounted for by the other and vice versa [23]. This multivariate method is largely applied to foods and beverages due to the ability to examine samples in their entirety, to untangle all the complicated interactions between the constituents and to understand their combined effects on the whole matrix. Note that the emphasis is on predicting the characteristics and not necessarily on explaining the underlying relationships between the variables [24].

Many authors have applied this method to study the relationships between sensory descriptors and volatile compounds in wines [23,25,26,27]. Regarding cider, PLS was used as a regression method, for example, to study the relation between the composition of the raw material (apples) and cider quality, to ascertain a general prediction rule for the stage of ripening of a set of apples selected for their technological suitability for cider making [28]. PLS model was also applied in a sensory study to predict bitter or non-bitter tastes in cider using six polyphenols as predictor variables [7]. Lobo et al. [29] applied PLS in the development and validation of prediction models to transform frequency signals obtained by Fourier transform infrared (FT-IR) spectroscopy into concentrations of the different components in the cider, which is useful for the routine analysis of samples.

Given the results of the previous studies, this study aimed to characterize hard-apple ciders from a chemical point of view and to study the relationship between the production process and customers’ sensory perceptions. Moreover, a sensory-dryness-prediction model to monitor the production process was proposed. To meet consumer demand and maintain consistent their perceptions of quality, a new marketable scale based on the model was set up, which informed customers about the kind style of cider they intended to purchase. This scale expressed the perceived dryness by considering taste and tactile attributes (astringency, bitterness, acidity and sweetness), which, in turn, were correlated with a set of selected chemical parameters considered as markers. In order to define the most effective chemical variables to predict dryness by an appropriate scale, the present study was articulated in several steps: (i) chemical and sensory characterization of ciders representative of the New York region’s production; (ii) study of the relation between their chemical and sensory profiles; (iii) analysis of the predictive capability of the chemical parameters by using a PLS analysis and the selection of the most effective parameters; (iv) model verification defined by a set of cider samples (calibration set) and by the dryness prediction of another set of cider samples (validation set); (v) comparison of the predicted values with the dryness scores attributed to the same samples, by the current scales, and by the evaluation of a panel of trained judges.

## 2. Materials and Methods

### 2.1. Cider Samples

Seventy-six commercial cider samples (0.75-L glass bottle) were collected from twenty-two facilities from the New York State area, representative of the different products and of the relevant production orchards (Table 1).

### 2.2. Chemical Analyses

#### 2.2.1. Chemical Standard Parameters

Residual sugar (glucose and fructose) and malic acid were measured using the enzymatic method on an automated chemical analyzer (Konelab 20XT, Thermo Fisher Scientific, Waltham, MA, USA) with test kits for discrete analyzers (Vintessential, Rowe Scientific, Adelaide, SA, Australia). The pH was analyzed using a standard pH probe (Ross Sure-Flow pH electrode, Thermo Orion, Thermo Fisher Scientific, Cleveland, OH, USA), and the titratable acidity was determined through potentiometric titration using 0.1 N NaOH and the Mettler Toledo T90 (1900 Polaris Parkway, Columbus, OH 43240, USA) instrument. Alcohol content was measured by gas chromatography using a Hewlett-Packard 5890 GC (Roseville, CA 95747, USA) combined with a flame-ionization detector (GC-FID) and according to official method of the Association of Official Analytical Chemists, or AOAC (International Method 983.13, Rockville, MD 20850, USA).

#### 2.2.2. Spectrophotometer Indices

Absorbances at 280 nm and 320 nm were measured using a 1-cm-path-length quartz cell and a UV-VIS spectrophotometer (Genesys™ 10S UV-VIS, Thermo Scientific, Cleveland, OH, USA). Samples were centrifuged before the analysis.

#### 2.2.3. Phenolic Analysis

The HPLC analysis was performed using an Agilent 1100 Series HPLC (Agilent Technologies, Santa Clara, CA, USA). Each sample was injected (5 μL) and chromatographically separated on a reverse-phase column LiChroCART 250-4 HPLC Cartridge LiChrospher 100 RP-18, endcapped (5 μm, 250 × 4 mm) (Agilent Technologies) and detected using a diode-array detector. Flow was 0.5 mL/min and the time run was 80 min. Solvent and gradients were set up according to Peng et al. [30]. Chromatograms were registered at 280 nm for the determination of tannins and hydroxybenzoic acids, at 316 nm for the determination of hydroxycinnamic acids and at 365 nm for the determination of flavanols. Calibration curves were obtained using the following standard compounds: gallic acid for hydroxybenzoic acids, caftaric acid for hydroxycinnamic acids, rutin for flavanols and (+)-catechin for tannins.

### 2.3. Sensory Analysis

The sensory tests were carried out in the late morning before lunch time, in a sensory-analysis laboratory equipped with individual cabins (temperature-controlled and combined natural/artificial light), designed in accordance with the ISO standard.

#### 2.3.1. Dryness Evaluation

Seventy-six samples were collected and submitted to a sensory analysis to evaluate the sensory “dryness” perception and the sensory evaluation of sweetness, acidity, astringency and bitterness. The panel consisted of 11 trained judges, who evaluated the cider samples in two separate sessions. The panel was trained and calibrated using the scale applied by the American Cider Association, Beer Judge Certification Program-BJCP, Great Lakes International Cider and Perry Competition (GLINTCAP) and New York Ci-der Association (NYCA), consisting of four categories of sweetness, defined according to the residual sugar content in cider, as follows: dry (<9 g/L), semi-dry (9–18 g/L), semi-sweet (18–45 g/L) and sweet (>45 g/L). In the first session, a category scale of 8 levels was set up. To anchor the dryness level in the perceived intensity of sweetness, three reference standards of sweetness (glucose in standard cider) were submitted to the judges: a total of 9 g/L of glucose, corresponding to a score of 2; 18 g/L, corresponding to a score of 4; and 45 g/L, corresponding to a score of 6. Based on these reference standards, the dryness levels were defined as follows: 0–2, corresponding to dry; 2.1–4, to semi-dry; 4.1–6, to semi-sweet; and 6.1–8. to sweet. According to this scale, the trained judges evaluated the cider samples’ dryness.

#### 2.3.2. Descriptive Analysis

In the second session, the evaluation of the samples was performed by using quantitative descriptive analysis (QDA). The panel leader trained the panelists in four 60-min training sessions. In the first session, the panelists were presented with a range of cider samples and invited to describe them through the evaluation of three taste attributes (sweetness, acidity, bitterness), and one mouthfeel descriptor (astringency). In the subsequent sessions, the judges were provided with a subset of samples and reference standards until they reached a consensus as to the attributes and the score-sheet sequencing. The references were prepared using a standard cider with addition of the compounds reported in Table 2 and corresponded to 6 (medium intensity) on an intensity-category scale, ranging from 1, on the left (not detected), to 10, on the right (very intense). The level of training of the panelists was checked by an individual evaluation of a subset of the samples and the statistical analysis of the data. The evaluation of the cider samples by the QDA was performed by serving them at a temperature of 12 °C in plastic test tubes. Each sample contained a constant volume of 30 mL of cider. Judges evaluated three taste attributes (sweetness, acidity, bitterness), and one mouthfeel descriptor (astringency). Samples were blind-tasted and coded with randomized three-digit numbers. Before each evaluation, panelists tasted the taste and mouthfeel standards, which were prepared following the same procedure as the panel training.

Panelists evaluated all samples in a total of 11 sessions. Each session featured 6 or 7 samples in duplicate, served across judges in a balanced incomplete block design. Given that some cider samples showed very different contents of residual sugar and alcohol and to avoid the influence of outliers, the sessions were organized in such a way that homogeneous group of samples for these compounds were presented. Judges were required to spit all the cider samples, wait 30 s between samples and cleanse their palates with water.

### 2.4. Statistics

Principal component analysis (PCA) and partial least-squares regression (PLS1) were performed using Unscrambler (ver. 10.3, CAMO Process AS, Bedford, MA, USA). Principal component analysis was performed before PLS models to examine any relevant and interpretable structures in the data and to detect outliers [31]. Principal component analysis was performed on the trained judges’ average scores attributed by QDA to the taste and tactile descriptors (sweetness, acidity, bitterness, astringency) after determination of their significance by ANOVA (Statgraphic Centurion Ver.XV, StatPoint Technologies, Warrenton, VA, USA). For the PLS1 analysis, the *x*-variables were represented by the selected chemical variables affecting the taste and tactile perception and, therefore, the dryness of the cider (residual sugars, alcohol, titratable acidity, malic acid, pH, tannins, polyphenol content, hydroxybenzoic and hydroxycinnamic acids, 280- and 320-nm absorbances); the *y*-variable was represented by the dryness scores. Cross-validation with randomization were used for prediction and the Nipals algorithm was applied.

## 3. Results and Discussion

### 3.1. Chemical and Sensory Characterization

The cider samples were submitted to chemical analysis for standard parameters such as alcohol content, pH, titratable acidity, malic acid content and residual sugar. Moreover, parameters such as the polyphenol content, hydroxybenzoic and hydroxycinnamic acids and absorbance at 280 nm and 320 nm were determined to define the polyphenol profiles of the samples. The results of the descriptive statistics are reported in Table 3. Samples were divided into two groups according to their chemical composition to form two homogeneous cider samples groups, in order to build and validate a prediction model. Table 3 shows that the two sample sets selected for the calibration and validation models presented similar chemical compositions. However, some exceptions were presented and highlighted by the maximum and minimum values for all the chemical parameters considered. In particular, the residual sugar content was found to be one of the parameters with the highest variability, as several samples showed very high values (i.e., 7689 mg/100 mL for calibration samples, and 17,059 mg/100 mL for validation samples) and, for this reason, they strongly affected the average value. The same trend, albeit less evident, was observed for other chemical parameters, such as the alcohol content, the malic acid and the related titratable acidity, as well as the polyphenol content. These chemical data indicated that the selected cider samples were substantially homogeneous, with some exceptions, which were suitably managed for both the evaluation and the model building.

This was confirmed by the PCA shown in Figure 1 (the KS2 and KS8 samples presented outliers and were removed from the data), which displays the distribution of the cider samples according to their chemical parameters (alcohol content, pH, titratable acidity, malic acid, hydroxybenzoic acids, hydroxycinnamic acids, polyphenol content, total tannins and residual sugar), their sensory attributes (sweetness, acidity, bitterness and astringency) and their sensory dryness.

The PCA results explained 52% of the variance and separated the cider samples along the first dimension (PC1 explained 36% of the variance) according to the chemical parameters of pH, hydroxycinnamic and hydroxybenzoic acids and alcohol on the right side of the graphic and residual sugar, malic acid, titratable acidity, polyphenols and total tannins on the left side. Furthermore, the samples were separated along the second dimension (PC2 explained 16% of the variance) according to the titratable acidity, malic acid, polyphenol content, hydroxycinnamic and hydroxybenzoic acids, total tannins, polyphenols and the spectrophotometric measurements abs 320 nm and 280 nm, which are correlated with polyphenol compounds, and pH on the opposite side. The sensory attributes of bitterness, astringency and acidity were related to the hydroxycinnamic acids and, albeit to a lesser extent, to the hydroxybenzoic acids, pH and alcohol content. Instead, the sweetness and sensory dryness were related to the residual sugar and, to a lesser extent, to the malic acid, titratable acidity, tannins and polyphenols. The distribution of the samples showed that most of them were concentrated between the fourth and the first quadrants and were related to the chemical parameters of pH, alcohol and hydroxycinnamic and hydroxybenzoic acids. The other samples were more spread out, with a different level of relationship with the other chemical parameters: RB1, LO2, LO1, MD3, KS7, BB1 and MD1 were strongly correlated with abs 280 nm, abs 320 nm, polyphenols, total tannins, titratable acidity and malic acid, while BB3, BB2, MD2, KS1, AO1, LO3, BB4, WE2, BH1, WE3, PC1, NP2, OH2 and AW1 were heavily related to chemical parameters such as residual sugar, titratable acidity and malic acid and the sensory attributes of sweetness and sensory dryness.

The sensory dryness was found to be strictly correlated to the sweetness perception, which, in turn, was directly correlated with the residual sugar content and negatively correlated with the pH, alcohol content, hydroxycinnamic and hydroxybenzoic acids and abs 280 nm. The relationship between the sensory attributes and chemical parameters demonstrated that sweetness hindered acidity and astringency, with some exceptions [32] and only when the residual sugar content was low (lower than ~15 g/L) but, overall, when hydroxycinnamic and hydroxybenzoic acids were high (generally more than about 50 mg/L for hydroxycinnamic acid and about 20 mg/L for hydroxybenzoic acid), the acidity and astringency became perceptible, which was also due to the synergism between these two oral perceptions [33]. The hydroxybenzoic and hydroxycinnamic acids, combined with high alcohol content, showed a greater ability to interfere with the sweetness perception than acid compounds, such as malic acid, and the titratable acidity. This confirmed that sweet taste suppressed astringency perception and that it was the only taste to be slightly or not affected by astringency [34]. On the other hand, these findings confirmed those of other studies, in which the ethanol significantly increased the perception of bitterness, while the acidic compounds decreased it [35]. The similarity in the trends of bitterness and astringency attributes could have been due, beyond the low sweetness level, to the fact that these two sensations were often induced by the same compounds, such as polyphenols [36,37]. Moreover, the correlation of the pH with the astringency and bitterness sensations seemed to confirm the findings of other studies on wine, in which the increase in the pH led to an increase in the perception of these attributes [38].

### 3.2. Definition of the Dryness Rating by Marketable Scales and Sensory Evaluation

In Table 4, the scores and the relative rating of the dryness attributed to the cider samples are reported as resulting from the application of four different scales: the International Riesling Scale (IRF) with and without pH correction, the New York Cider Association (NYCA) scale and scores attributed by the Quantitative Descriptive Analysis (QDA). Although the IRF rating demonstrated that the ratio between the residual sugar and the malic acid had to be corrected as a function of the pH value, the IRF scores without pH correction were also considered as they represented the base value for the calculation of the NYCA-scale rating (see Table 5 and Table 6).

The values attributed to the ciders by the four scales were quite similar for most of the samples, although some relevant exceptions were detected (Table 4).

The IRF scale without pH adjustment expressed the dryness as the ratio between the residual sugar and the titratable acidity (expressed as the malic acid content), assuming that the perceived intensities of sweet and sour tastes were different when both tastes were in the same mixture. Mixture suppression is a phenomenon in which the perceived intensity of two tastes in a mixture is lower than if they were unmixed, at the same concentration level [39]. Generally, sourness is suppressed by sweetness with a stable pattern, depending on the levels of both components. Previous studies showed that the overall perceived intensity was the result of “perceptual additivity,” i.e., the additivity of the taste intensity, that is, the perceived taste within the mixture, rather than the “stimulus intensity” [40,41,42,43]. Moreover, the influence of the intensity concentration on the taste interaction was modeled according to a psychophysical function, in which the ratio between different intensity perceptions is different at low, medium, and high levels of perception [44]. This function can also be applied to sweetness, the most heavily related taste to perceived dryness (see PCA Figure 1), and it is generally considered an especially effective suppressor of sourness, which is also affected by the alcohol, which generally enhances the perception of sweetness [32]. Therefore, the ratio between the concentrations of the two stimuli (residual sugar and malic acid), given that their intensity perception in a complex mixture such as cider was unknown, represented a poor predictor of the overall perceived intensity of the related taste. The IRF scale presented another critical point in relation to the calculation modality of the rating: when the malic acid concentration was low or very low (a fraction of mg per liter), the ratio and, hence, the scores, became very high (WW5, SH3, WW1, PF1, WW4, SH5 and TC3), although the residual sugar of the sample was not relevant in terms of its perception (Table 4). For example, the sample WW5 had a ratio of 5, corresponding to the IRF rating of “sweet,” resulting from the ratios of 5 mg/100 mL of residual sugar and 0.001 g/L of malic acid. The score for the same sample obtained by the sensory evaluation of dryness, given the concentration of the residual sugar (5 mg/100 mL), was probably more realistic and noticeably lower (2.9), corresponding to a “semi-dry” rating. A similar situation occurred with the sample SH3, with a ratio of 9, corresponding to the rating of “sweet,” resulting from the ratios of 90 mg/100 mL of residual sugar and 0.001 g/L of malic acid, presenting a score obtained by the sensory evaluation of dryness of 2, which is equivalent to a “dry” rating. The most evident case was that of the sample SH5, with a residual sugar level of 178 mg/100 mL and a similarly low content of malic acid (0.001 g/L), which presented a very high IRF value of 178, corresponding to a “sweet” rating, while the sensory dryness score attributed by the panel of trained judges was 2.7, corresponding to “semi-dry”.

The IRF scale with pH correction demonstrated that the ratio between the residual sugar and the malic acid had to be adjusted as a function of the pH of the sample (Table 5), assuming that the higher the pH, the higher the dryness perception and that other acid compounds than malic acid significantly contributed to the perceived acidity. The studies on this topic provided conflicting results. Stone et al. [45] found that in an aqueous solution, reducing the pH from 5.8 to 4.0 had little effect on the sweetness of glucose and fructose, but a reduction from 4.0 to 2.7 caused a 50% reduction in the relative sweetness.

The study by Stevens [46] found that weak concentrations of citric acid (e.g., below 10^−3^ M) had little effect on sucrose thresholds, while higher concentrations (2 × 10^−3^ M and above) slightly elevated sucrose thresholds; the pH values were not reported. Schimann et al. [47] evaluated the effect of pH on sweetness, at five different levels (pH 3.0, 4.0, 5.0, 6.0 and 7.0) in aqueous solutions of HCl or NaOH. No significant changes in perceived sweetness were found over a pH range from 3 to 7, but the acidity and astringency perceptions increased with decreasing pH values. The latter finding could in some way support the IRF scale with the pH correction, given that the perceived dryness was a sensation which could contribute to both the perception of sweetness and to the acidity. Compared to the others, the scores obtained by this scale were found to be the higher for almost all the samples, indicating that the pH adjustment had a strong effect and noticeably increased the rating, however it seems to have led to oversized values, since almost none the resulting scores were confirmed by those of the panel of trained judges.

The NYCA scale (Table 4) was used to calculate the perceived dryness according to the ratio of the residual sugar content to the malic acid (corresponding to IRF score), adjusted as a function of the tannin concentration. This scale assumed that tannins were the main compounds responsible for astringency and, hence, that they are able to decrease the sweetness perception. Polyphenols (tannins and hydroxybenzoic and hydroxycinnamic acids) play an important role in cider quality, since they are responsible for its color and the balance of bitterness and astringency, which defines the “overall mouthfeel” of ciders. Several studies confirmed that astringency depends mainly on tannin content [19,20,21,35] and that its perception can be enhanced or depressed as a function of different factors. However, there were conflicting results regarding the effects of sweetness on astringency. Lyman et al. [48] found that the addition of 0.5 M of sucrose to 1000 mg/L tannic acid solution reduced the perception of dryness, while Brannan et al. [34]’s results showed that in a model solution of alum and tannin at low concentrations, astringency was not affected by sweetness, while at high concentrations, sweet taste suppressed astringency perception. The authors concluded that sweetness was the only taste to be slightly or not affected by astringency.

Studies on pH state that in general, the higher the pH, the lower the perceived astringency [49,50,51]. Furthermore, studies on the effect of total acidity on the perceived astringency have hypothesized that it was not significant [51,52] and that the astringency elicited by acids was a function of pH and not of acid concentration [53,54].

The NYCA scale provided the values that must be subtracted from the ratio between the residual sugar and the malic acid, in relation to the concentration of tannins (Table 6). This scale, did not change the calculated IRF-dryness score when the tannin concentration was lower than 500 mg/L and decreased it for higher values as a function of their concentration, up to a maximum of ¾ of a unit, for values equal to or more than 1000 mg/L tannins. Although it seemed to have a good relationship with the perceived dryness evaluated by the panel of trained judges, when there was a very high tannin content, the scale appeared to lose its ability to detect their effect on the perception. This was the case with the sample MD1, which had an IRF value of 4.009, corresponding to a “sweet” rating but, given the high content of tannins (5000 mg/L), a final value on the NYCA scale was 3.32, which corresponded to a “semi-sweet” rating. In fact, the perceived dryness assigned by the scale determined by the panel of trained judges was clearly lower (3.6), corresponding to the “semi-dry” rating. This can be explained by the fact that the very high content of tannins had a strong effect on the perceived dryness, while the NYCA tannin correction affected the score values both at 1000 and 5000 mg/L of tannins (Table 6) in the same way.

The sensory evaluation of the cider samples by the QDA method confirmed that the dryness rating was directly correlated with the perceived sweetness (Table 4). Moreover, given the mixture’s effect [39] on the perceived sweetness represented by the results of the interactions among the different tastes and tactile stimuli contained in the mixture, QDA is the most effective method for providing the final perception of the overall intensity according to the tastes perceived after any and all peripheral and central interactions occur [40,42,43].

According to these results, it is possible to highlight that the marketable scales discussed here, despite the improvements introduced (see the NYCA scale), take into consideration the different chemical compounds that directly influence the perception of dryness without consider their interactions. To better correlate the scores with the actual perception of dryness, including the interaction effects of the different tastes and tactile sensations, a multivariate approach can be exploited.

### 3.3. Dryness Prediction by Chemical Parameters: Set-Up of the Model and Validation

To define a model for the prediction of the dryness, it was necessary to focus on the relation between the perceived dryness and the chemical parameters related to its perception. Given the studies on this topic, several chemical parameters were selected as a function of their direct or indirect influence on taste and tactile perception: residual sugars and alcohol for sweetness; titratable acidity, malic acids and pH for acidity; and tannins, polyphenol content, hydroxybenzoic and hydroxycinnamic acids and 280-nm and 320-nm absorbances for astringency and bitterness.

Partial least squares (PLS1) regression was applied to compare the dryness scores with the chemical compounds of a set of cider samples (calibration set), with the goal of setting up a model to predict the dryness of a new set of ciders (validation set). When the first model was set up, the Hotelling T^2^ Test showed that there was an outlier (KS8) and a new model was set up without that sample.

The resulting model is reported in Figure 2a, while Figure 2b reports the Bw regression coefficients. The values of explained variance (24%, 20% for Factor-1 and 71%, 10% for Factor-2) were at their maximum after Factor-3, so these factors were considered for the prediction. The predicted-versus-reference plot (Figure 2a) showed that it was a good-quality model (R^2^ = 0.909; RMSEC = 0.472; SEC = 0.478). The predictions made during the calibration were checked against the predicted-versus-measured plot to examine the ability of the single chemical variables to model the dryness, but their importance was better summarized as weighted regression coefficients (Bw) (Figure 2b). The Bw coefficients made it possible to explain their weight and the quality of the relationship with the dryness parameter. Moreover, in order to improve the model’s efficiency, it was more convenient to select the most predictive chemical variables, i.e., those whose regression coefficients had estimated uncertainty limits within the 95% confidence interval (Figure 2b, bars marketed with grid). The variables with uncertainty limits crossing the zero line were not significant even if they presented large regression coefficients because the estimated uncertainty was consistent with the relationship between such variables and Y (dryness), since only some of the samples spanned the range. The regression coefficients showed that the residual sugar, malic acid, abs 320 nm and hydroxybenzoic acid had a direct relationship with the predicted dryness, while the total tannins, polyphenols, hydroxycinnamic acid, abs 280 nm, pH and alcohol content were negatively correlated. The uncertainty-test limits and the Bw coefficients indicated that the residual sugar, hydroxycinnamic acid and abs 280 nm were the chemical variables that were able to significantly predict the dryness, whose predicted values are shown in Table 7. The defined model was tested by the prediction of the dryness scores of a second set of ciders, the validation set, after their transformation into the corresponding rating values. The predicted ratings were compared to the sensory dryness ratings obtained through the QDA analysis and the NYCA scale. The IRF scale was not considered.

The values predicted by the model set up based on all the chemical variables (Table 7—model #1: residual sugar, titratable acidity, malic acid, tannins, polyphenols, hydroxycinnamic and hydroxybenzoic acids, abs 280, abs 320, alcohol content and pH), were found to be more similar to the evaluated sensory dryness than the NYCA scale. In fact, the NYCA scale predicted different values with respect to the evaluated dryness for twenty samples, three of which (SH5, SH11 and TC3) had two different ratings, while the predicted model’s values showed different ratings for 18 samples, none of which had more than one level. A large deviation from the predicted value was also detected for the sample KS2, but this can be explained by the very high content of residual sugar (177 g/L), which made it an outlier and, hence, difficult to predict with precision within of this category of cider.

The model set up by the important variables (Factor-1: 50%, 39%; Factor-2: 75%, 6%; R^2^ = 0.901; RMSEC = 0.490; SEC = 0.497), which were the variables whose regression coefficients had estimated uncertainty limits within the 95% confidence interval (Figure 3: residual sugar, pH and abs 280 nm), showed an even better performance, as 14 rating levels were found to be different from the sensory dryness scores, versus 20 on the NYCA scale and 18 from the model defined by all the chemical variables (Table 7; model #2). Although the samples SH11, RB1 and RB3 presented a larger range of variation, the dryness values predicted by the defined model suggest that it was possible to obtain a good level of prediction by using three defined chemical variables for the tested cider, i.e., the residual sugar, hydroxycinnamic acid and abs 280 nm.

#### Selected-Variables Model

Given the purposes of this study, which were to set up a model to predict dryness based on appropriate chemical parameters and to explore the possibility of refine the model itself by making it easily applicable in the cider-production process, the choice of chemical variables was considered in light of their readiness for use in an ordinary laboratory analysis of a production company.

Residual sugar is a common, easy and inexpensive parameter, determined by a specific enzymatic method and spectrophotometric analysis. The polyphenols presented some difficulties, as they were measured by a HPLC method, a very expensive instrument that can only be handled by specialized technicians; moreover, the polyphenols included a pool of chemical compounds (which also included hydroxybenzoic and hydroxycinnamic acids), which had a different kind and level of influence on the perception of sweetness. The measurement of abs 280 nm is in fact, an easy and widely used method determinate the polyphenols content, together with abs 320 nm, correlated to hydroxybenzoic and hydroxycinnamic acids [55]. For these reasons, this study was performed to test the predictive power of easily measurable parameters correlated with polyphenol contents, such as 280-nm and 320-nm absorbance, and the chemical compounds and parameters influencing sweetness perception beyond residual sugar, such as pH and titratable acidity, which are usually determined in the quality-control process and correlated with the perception of sweetness.

The new model is reported in Figure 4. The explained variance (35%, 73% for Factor-1 and 28%, 10% for Factor-2) was at its maximum after Factor-3, so these factors were considered for the prediction. The predicted-versus-reference plot (Figure 4a), with R^2^ equal to 0.857, RMSEC to 0.441 and SEC to 0.447, showed a good level of prediction. The uncertainty test and Bw coefficients made it possible to identify the chemical variables that significantly predicted the dryness, i.e., the residual sugar was positively correlated and the pH and titratable acidity were negatively correlated with the dryness. These results confirmed the relation, evidenced by the PCA (Figure 1), between acidity (together with astringency), and sweetness in the perception of the dryness.

The model set up based on the selected chemical variables (model #3—residual sugar, titratable acidity, abs 280, abs 320 and pH) was tested in the same way as the others, by predicting the dryness values of the validation set and comparing the resulting ratings with those calculated by all the variables, the important variables model and the NYCA scale. The comparison made it possible to conclude that the model was more similar to the evaluated sensory dryness than the NYCA scale, and even more effective than the important-variables-based model itself. In fact, the ratings predicted by model #3 showed a different rating from the sensory evaluated dryness for just 11 samples, none of which for more than one level, compared with the 18 samples with different ratings on the NYCA scale and the 14 samples in the important-variables model. Using the latter model, the sample SH11 also demonstrated better predictability, while the large range of values of the sample KS2 was confirmed. The model equation used to calculate the dryness score is as follows (Equation (1)):(1)Dryness (y)=5.928−0.929×pH+0.0687×Titratable acidity−0.425×Abs 280 nm−0.0395×Abs 320 nm+0.0005243×Residual sugar

## 4. Conclusions

The traditional scales applied so far (IRF and NYCA scales) to define the dryness of cider cannot comprehend the complexity of the relationship between chemical and sensory profiles when using a univariate approach, in which single compounds are considered separately. On the other hand, sensory evaluation is complex, expensive and time-consuming and cannot be used as an ordinary tool to define marketable product parameters. Given that the relationships between sensory perception and chemical composition are complex, a predictive model for dryness using the multivariate method of PLS was applied, using several chemical variables of cider that are correlated with dryness. The PLS method is an important tool to study the relationship between chemical and sensory data, with important practical applications. The PLS methodology makes it possible to untangle the complicated interactions between chemical constituents to understand their combined effects on perceptions and to explore the relationships between chemical compounds and the sensory perception of dryness.

The determination of cider dryness by using chemical-composition data is challenging because the chemical interactions are complex. Studies that account for this complexity using statistical correlation could be of interest for this purpose.

However, further research is needed, as this topic has not yet been completely explored. It is important to note that to obtain an efficient and effective predictive model and, given the large variability of several chemical parameters, specific studies on different kinds of cider are needed. As with every predictive model, the extension and application of this predictive model over time will be necessary for its optimization.

## Figures and Tables

**Figure 1 foods-12-02191-f001:**
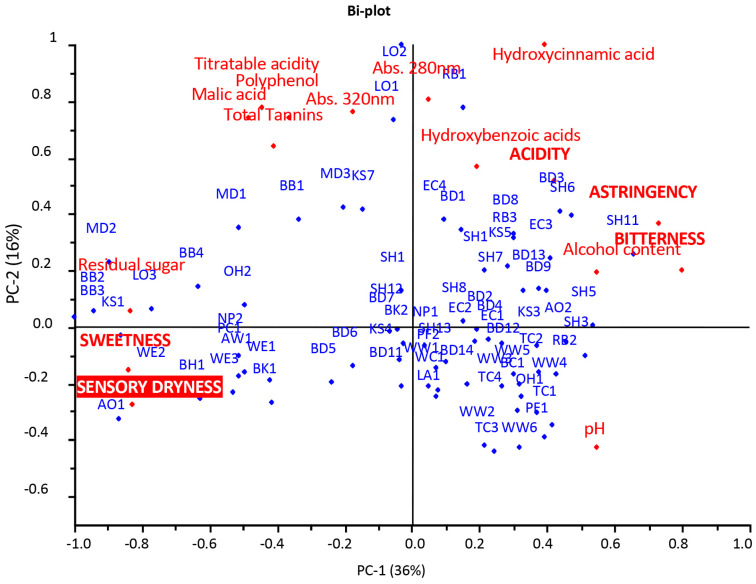
Principal component analysis (PCA): scores-and-loadings bi-plot of chemical parameters, sensory attributes and dryness.

**Figure 2 foods-12-02191-f002:**
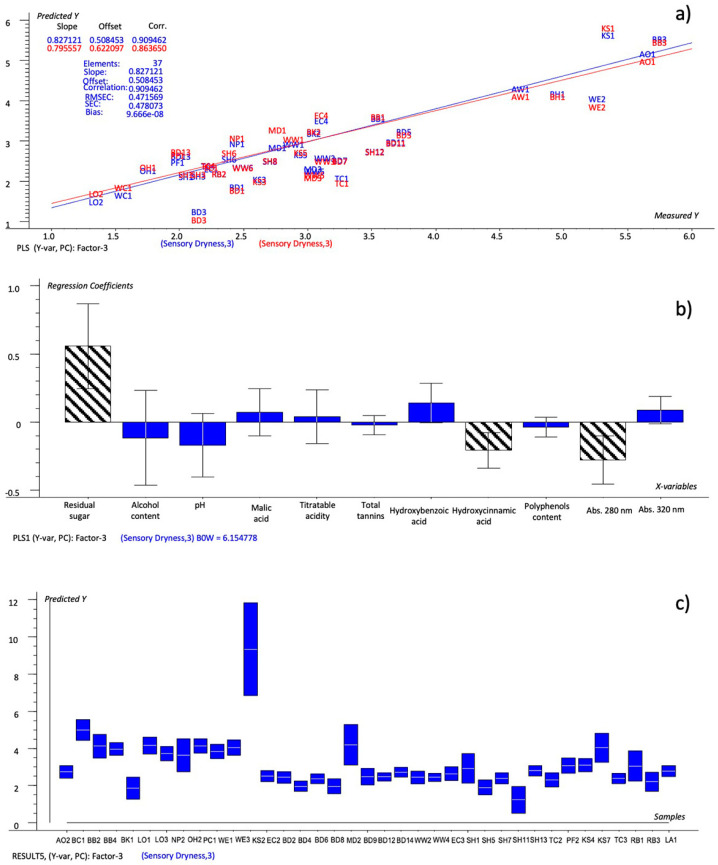
PLS 1 plots: (**a**) Prediction model (sensory dryness predicted by all chemical variables); (**b**) weighted regression coefficients—Bw (important variables reported as black-striped bars) with estimated 95% confidence intervals; (**c**) predicted of dryness values for cider samples by the prediction model with uncertainty limits.

**Figure 3 foods-12-02191-f003:**
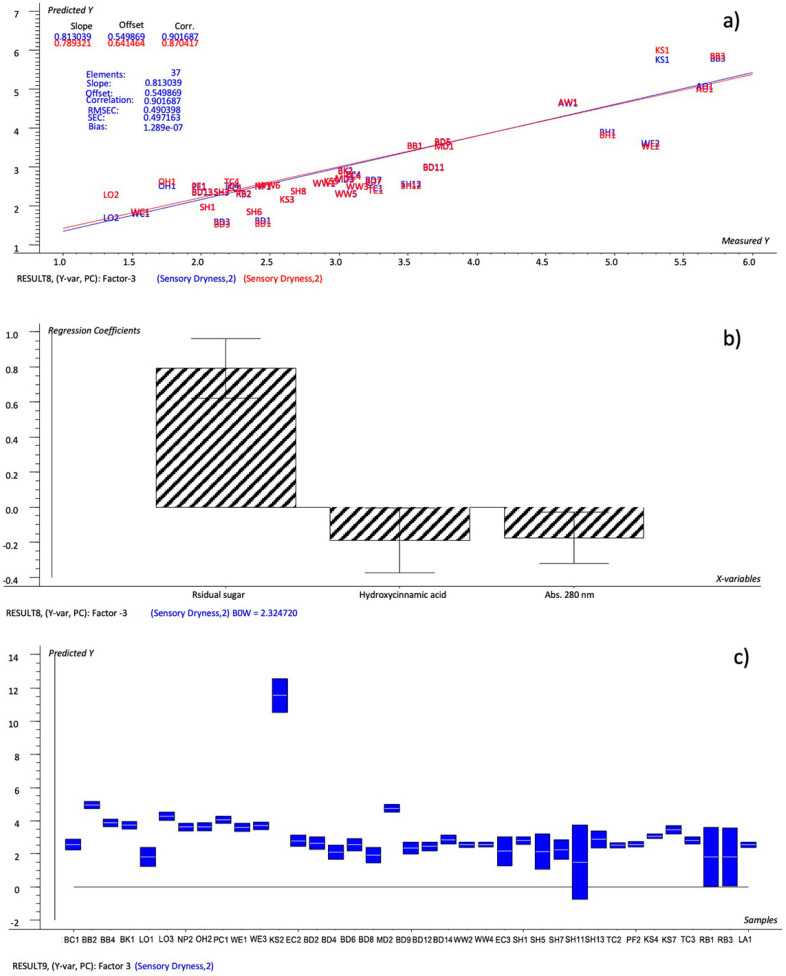
PLS 1 plots: (**a**) Prediction model (sensory dryness predicted by important chemical variables); (**b**) weighted regression coefficients—Bw (important variables reported as black-striped bars) with estimated 95% confidence intervals; (**c**) dryness values predicted for cider samples by the prediction model with uncertainty limits.

**Figure 4 foods-12-02191-f004:**
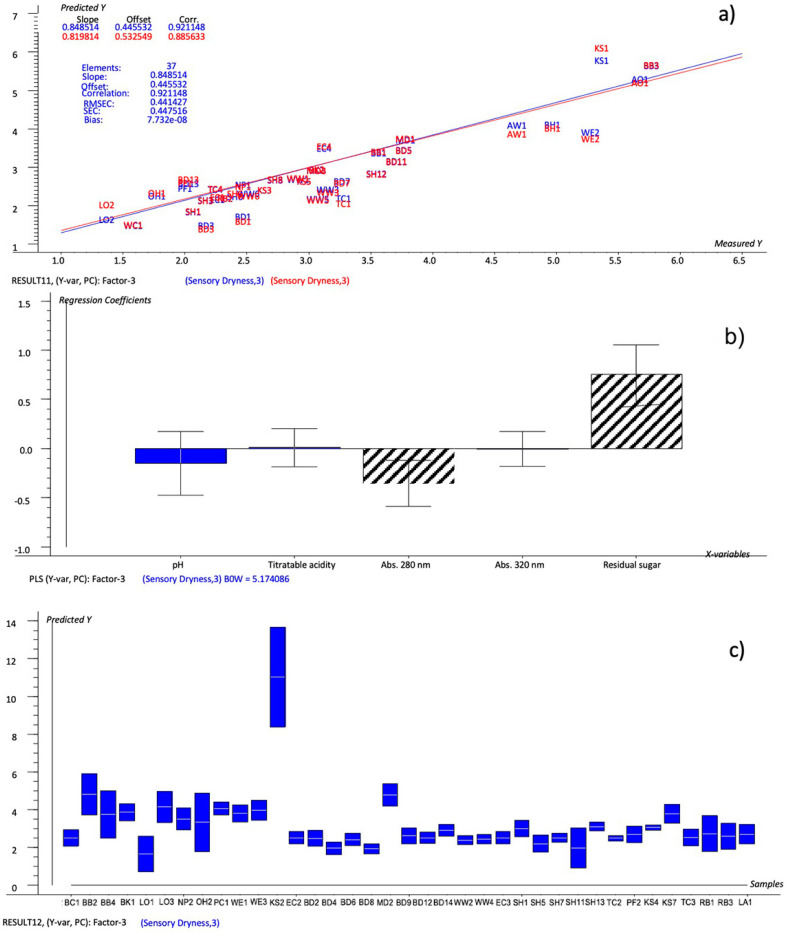
PLS 1 plots: (**a**) Prediction model (sensory dryness predicted by selected chemical variables); (**b**) weighted regression coefficients—Bw (important variables reported as black stripes bars) with estimated 95% confidence intervals; (**c**) dryness values predicted for cider samples by the prediction model with uncertainty limits.

**Table 1 foods-12-02191-t001:** Cider facilities, codes and number of samples collected from every producer.

Cider Facility	Cider-Facility Code	Number of Samples
Angry Orchard	AO	2
Applewood Winery	AW	1
Bad Seed Cider Co.	BC	1
Blackbird Cider Works	BB	4
Blackman Homestead	BH	1
Brooklyn Cider House	BK	2
Leonard Oakes Estate Winery	LO	3
Nine Pin	NP	2
Orchard Hill	OH	2
Phonograph Cellars	PC	1
South Hill Cider	SH	10
Wayside Cider	WC	1
Wolffer Estate	WE	3
Kite and String	KS	7
Eve’s Cidery	EC	4
Black Diamond	BD	13
Merchants Daughter	MD	3
Westwind Orchard	WW	6
Treasury Cider	D23	4
Pennings Farm Cidery	PF	2
Redbyrd Orchard Cider	RB	3
Little Apple Cidery	LA	1

**Table 2 foods-12-02191-t002:** Taste and mouthfeel standard recipes: compounds added to the standard cider and their relative concentrations, corresponding to 6 on the intensity scale.

Taste and Mouthfeel Standards Recipe	Compound	Dose (g/L)	Product
Acidity	citric acid	7	Toscolapi (Italy) enological use
Sweetness	cane sugar	12	Italia Zuccheri (Italy)
Bitterness	caffeine	1	Sigma & Aldrich (Saint Louis, MO, USA), food grade
Astringency	alum	7	A.C.E.F. (Italy)

**Table 3 foods-12-02191-t003:** Concentration ranges and absorbances (max, min, median, lower quartile Q1 (25°), upper quartile Q3 (75°) and average) of calibration and validation cider samples.

	Alcohol Content (% *v*/*v*)	pH	Malic Acid (g/L)	Titratable acidity (g/L Malic Acid)	Residual Sugar (mg/100 mL)	Polyphenols Content (mg/L)	Hydroxybenzoic Acids (mg/L)	Hydroxycinnamic Acids (mg/L)	Abs. 320 nm	Abs. 280 nm
** *Calibration samples* **										
**Max**	20.22	3.98	0.83	0.93	7689.00	5399.0	109.7	398.0	0.3758	4.5855
**Min**	5.03	3.26	0.00	0.25	1.00	50.4	15.0	7.0	0.0120	0.0279
**Median**	7.28	3.62	0.47	0.56	361.83	435.7	24.5	76.5	0.0760	0.2329
**Lower quartile (Q1)**	6.69	3.50	0.22	0.43	8.08	210.5	20.8	27.0	0.0523	0.1207
**Upper quartile (Q3)**	8.16	3.82	0.61	0.69	1738.67	702.1	28.8	139.3	0.1492	0.3858
**Average**	7.73	3.64	0.40	0.56	1248.51	824.1	28.9	101.2	0.1117	0.5940
** *Validation samples* **										
**Max**	19.03	3.99	1.45	1.62	17,059.00	5356.2	94.3	390.3	0.4266	3.2950
**Min**	5.67	3.27	0.00	0.33	1.00	62.0	16.0	15.0	0.0248	0.0476
**Median**	7.68	3.63	0.48	0.55	691.17	409.4	24.0	75.7	0.0812	0.2130
**Lower quartile (Q1)**	6.79	3.54	0.37	0.48	11.67	258.0	20.0	30.3	0.0558	0.1028
**Upper quartile (Q3)**	8.18	3.68	0.61	0.69	1953.25	744.4	31.3	126.3	0.1730	0.3730
**Average**	7.98	3.61	0.50	0.61	1758.83	793.8	31.4	101.5	0.1313	0.4811

**Table 4 foods-12-02191-t004:** Scores and ratings of the cider samples: IRF scale with and without pH adjustment; NYCA scale; sensory dryness by the panel of trained judges.

	IRF Scale Scores (Residual Sugar/Malic Acid)	IRF Rating without pH Correction	IRF Rating with pH Correction	NYCA Scale Scores (without pH Correction) IRF with Tannin Correction	NYCA Rating Rating (No pH Correction-IRF with Tannin Correction)	Sensory Dryness Scores	Sensory Rating
AO1	10.049	sweet	sweet	10.050	sweet	5.5	semi-sweet
AW1	9.259	sweet	sweet	8.510	sweet	4.5	semi-sweet
BB1	3.814	semi-sweet	sweet	3.060	semi-sweet	3.4	semi-sweet
BB3	9.534	sweet	sweet	8.780	sweet	5.6	semi-sweet
BH1	3.979	semi-sweet	sweet	3.980	semi-sweet	4.8	semi-sweet
BK2	1.213	semi-dry	semi-sweet	1.210	semi-dry	2.9	semi-dry
LO2	2.587	semi-sweet	sweet	1.837	semi-dry	1.2	dry
NP1	0.020	dry	semi-sweet	0.020	dry	2.3	semi-dry
OH1	0.034	dry	semi-sweet	0.100	dry	1.6	dry
SH1	0.009	dry	semi-sweet	0.002	dry	1.9	dry
WC1	0.002	dry	semi-sweet	0.002	dry	1.4	dry
WE2	3.211	semi-sweet	semi-dry	3.211	semi-sweet	5.1	semi-sweet
KS1	7.265	sweet	sweet	7.260	sweet	5.2	semi-sweet
EC1	0.920	dry	semi-sweet	0.920	dry	2.1	semi-dry
BD1	0.020	dry	semi-dry	0.020	dry	2.3	semi-dry
BD3	0.460	dry	semi-sweet	0.460	dry	2.0	dry
BD5	4.330	sweet	sweet	4.330	sweet	3.6	semi-dry
BD7	2.640	semi-sweet	semi-sweet	2.640	semi-sweet	3.1	semi-dry
MD1	4.069	sweet	sweet	3.320	semi-sweet	3.6	semi-dry
MD3	1.008	semi-dry	semi-sweet	0.258	dry	2.9	semi-dry
BD11	4.772	sweet	sweet	4.522	sweet	3.5	semi-dry
BD13	0.048	dry	semi-dry	0.047	dry	1.8	dry
WW1	11.000	sweet	sweet	11.000	sweet	2.6	semi-dry
WW3	1.000	dry	semi-sweet	1.000	dry	3.0	semi-dry
WW5	5.000	sweet	sweet	4.740	sweet	2.9	semi-dry
EC4	4.137	sweet	sweet	3.640	semi-sweet	3.0	semi-dry
SH3	9.000	sweet	semi-sweet	8.250	sweet	2.0	dry
SH6	0.622	dry	semi-dry	0.122	dry	2.2	semi-dry
SH8	0.521	dry	semi-dry	0.521	dry	2.6	semi-dry
SH12	1.657	semi-dry	semi-sweet	1.450	semi-dry	3.4	semi-dry
TC1	0.533	dry	semi-dry	0.533	dry	2.3	semi-dry
PF1	7.333	sweet	sweet	7.333	sweet	1.8	dry
KS3	0.022	dry	semi-dry	0.001	dry	2.5	semi-dry
KS5	0.622	dry	semi-dry	0.622	dry	2.8	semi-dry
KS8	20.449	sweet	sweet	20.450	sweet	6.3	sweet
TC4	0.037	dry	semi-dry	0.012	dry	2.8	semi-dry
RB2	0.140	dry	semi-sweet	0.140	dry	2.2	semi-dry
WW6	1.000	dry	semi-sweet	1.000	dry	2.3	semi-dry
AO2	0.002	dry	semi-sweet	0.002	dry	2.0	dry
BC1	0.002	dry	semi-sweet	0.002	dry	1.5	dry
BB2	6.700	sweet	sweet	5.950	sweet	5.2	semi-sweet
BB4	3.744	semi-sweet	sweet	2.990	semi-sweet	4.4	semi-sweet
BK1	3.994	semi-sweet	sweet	3.994	semi-sweet	3.5	semi-dry
LO1	1.335	semi-dry	semi-sweet	0.580	dry	2.3	semi-dry
LO3	5.827	sweet	sweet	5.080	sweet	4.4	semi-sweet
NP2	3.722	semi-sweet	sweet	2.970	semi-sweet	4.3	semi-sweet
OH2	4.617	sweet	sweet	3.870	semi-sweet	4.1	semi-sweet
PC1	3.934	semi-sweet	sweet	3.934	semi-sweet	3.8	semi-dry
WE1	3.080	semi-sweet	sweet	3.080	semi-sweet	3.9	semi-dry
WE3	3.353	semi-sweet	sweet	3.353	semi-sweet	4.4	semi-sweet
KS2	11.789	sweet	sweet	11.789	sweet	7.0	semi-sweet
EC2	1.850	semi-dry	semi-sweet	1.850	semi-dry	2.3	semi-dry
BD2	2.195	semi-dry	semi-sweet	2.195	semi-dry	2.2	semi-dry
BD4	0.004	dry	semi-dry	0.004	dry	1.8	dry
BD6	1.670	semi-dry	semi-sweet	1.670	semi-dry	3.8	semi-dry
BD8	0.019	dry	semi-sweet	0.019	dry	1.7	dry
MD2	8.261	sweet	sweet	7.510	sweet	5.8	semi-sweet
BD9	0.032	dry	semi-sweet	0.032	dry	1.8	dry
BD12	0.051	dry	semi-sweet	0.051	dry	2.4	semi-dry
BD14	4.949	sweet	sweet	4.949	sweet	3.2	semi-dry
WW2	1.000	dry	semi-sweet	1.000	dry	2.6	semi-dry
WW4	4.000	semi-sweet	sweet	4.000	semi-sweet	2.6	semi-dry
EC3	0.030	dry	semi-sweet	0.030	dry	2.0	dry
SH1	0.766	dry	semi-sweet	0.766	dry	3.7	semi-dry
SH5	178.000	sweet	sweet	178.000	sweet	2.7	semi-dry
SH7	0.011	dry	semi-sweet	0.011	dry	1.9	dry
SH11	9.554	sweet	sweet	9.554	sweet	1.8	dry
SH13	3.299	semi-sweet	sweet	3.299	semi-sweet	3.3	semi-dry
TC2	0.047	dry	semi-sweet	0.047	dry	2.0	dry
PF2	0.019	dry	semi-sweet	0.019	dry	2.0	dry
KS4	1.660	semi-dry	semi-sweet	1.660	semi-dry	3.7	semi-dry
KS7	1.724	semi-dry	semi-dry	1.724	semi-dry	2.9	semi-dry
TC3	11.931	sweet	sweet	11.931	sweet	3.1	semi-dry
RB1	0.179	dry	semi-dry	0.179	dry	2.4	semi-dry
RB3	0.193	dry	semi-sweet	0.193	dry	2.1	semi-dry
LA1	0.011	dry	semi-sweet	0.011	dry	3.1	semi-dry

**Table 5 foods-12-02191-t005:** IRF scale: technical guidelines for the calculation of the cider samples’ dryness.

IRF Scale—Technical Guideline					
	Sugar-to-Acid Ratio	pH		pH	Shift Due to pH
**DRY**	<1.0	3.1 to 3.2	if	= or >3.3	Medium Dry
				3.5 or >	Medium Sweet
**MEDIUM DRY**	1.0 to 2.0			= or >3.3	Medium Sweet
				< or =2.9	Dry
**MEDIUM SWEET**	2.1 to 4.0			= or >3.3	Sweet
				< or =2.9	Medium Dry
				< or =2.8	Dry
**SWEET**	>4.0			< or =2.9	Medium Sweet
				< or =2.8	Medium Dry

**Table 6 foods-12-02191-t006:** IRF tannin adjustment for the calculation of the NYCA rating (RS: residual sugar; MA: malic acid).

TANNIN CORRECTION	
<500 ppm	The numerical RS/MA remains unadjusted
501 to 750	The RS/MA is reduced by ¼ of the unit
751 to 1000	The RS/MA is reduced by ½ of the unit
>1000 ppm	The RS/MA is reduced by ¾ of the unit

**Table 7 foods-12-02191-t007:** Predicted dryness rating by the PLS1 models (model #1: all chemical variables; model #2: important chemical variables; model #3: selected chemical variables) and comparison with NYCA and QDA ratings.

Cider Samples Validation Set	NYCA Scale	Sensory Dryness by QDA	Predicted Dryness Rating by All Chemical Variables (Model #1)	Predicted Dryness Rating by Important Chemical Variables (Model #2)	Predicted Dryness Rating by Selected Chemical Variables (Model #3)
AO2	dry	dry	semi-dry	semi-dry	semi-dry
BC1	dry	dry	semi-dry	semi-dry	semi-dry
BB2	sweet	semi-sweet	semi-sweet	semi-sweet	semi-sweet
BB4	semi-sweet	semi-sweet	semi-dry	semi-dry	semi-sweet
BK1	semi-sweet	semi-dry	semi-sweet	semi-dry	semi-dry
LO1	dry	semi-dry	dry	dry	dry
LO3	sweet	semi-sweet	semi-dry	semi-sweet	semi-sweet
NP2	semi-sweet	semi-sweet	semi-dry	semi-dry	semi-sweet
OH2	semi-sweet	semi-sweet	semi-dry	semi-dry	semi-dry
PC1	semi-sweet	semi-dry	semi-sweet	semi-sweet	semi-sweet
WE1	semi-sweet	semi-dry	semi-dry	semi-dry	semi-dry
WE3	semi-sweet	semi-sweet	semi-sweet	semi-dry	semi-sweet
KS2	sweet	semi-sweet	sweet	sweet	sweet
EC2	semi-dry	semi-dry	semi-dry	semi-dry	semi-dry
BD2	semi-dry	semi-dry	semi-dry	semi-dry	semi-dry
BD4	dry	dry	semidry	dry	dry
BD6	semi-dry	semi-dry	semi-dry	semi-dry	semi-dry
BD8	dry	dry	dry	dry	dry
MD2	sweet	semi-sweet	semi-sweet	semi-sweet	semi-sweet
BD9	dry	dry	semi-dry	semi-dry	semidry
BD12	dry	semi-dry	semi-dry	semi-dry	semi-dry
BD14	sweet	semi-dry	semi-dry	semi-dry	semi-dry
WW2	dry	semi-dry	semi-dry	semi-dry	semi-dry
WW4	semi-sweet	semi-dry	semi-dry	semi-dry	semi-dry
EC3	dry	dry	semi-dry	semi-dry	semi-dry
SH1	dry	semi-dry	semi-dry	semi-dry	semi-dry
SH5	sweet	semi-dry	dry	semi-dry	semi-dry
SH7	dry	dry	semi-dry	semi-dry	semi-dry
SH11	sweet	dry	sweet	dry	dry
SH13	semi-sweet	semi-dry	semi-dry	semi-dry	semi-dry
TC2	dry	dry	semi-dry	semi-dry	semi-dry
PF2	dry	dry	semi-dry	semi-dry	semi-dry
KS4	semi-dry	semi-dry	semi-dry	semi-dry	semi-dry
KS7	semi-dry	semi-dry	semi-dry	semi-dry	semi-sweet
TC3	sweet	semi-dry	semi-dry	semi-dry	semi-dry
RB1	dry	semi-dry	semi-dry	semi-dry	semi-dry
RB3	dry	semi-dry	semi-dry	semi-dry	semi-dry
LA1	dry	semi-dry	semi-dry	semi-dry	semi-dry

## Data Availability

The data used to support the findings of this study can be made available by the corresponding author upon request.

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
