# Peer review of "Chemical Characterization, Sensory Definition and Prediction Model of the Cider Dryness from New York State Apples"

_foods, 2023, doi:10.3390/foods12112191_

Round 1

Reviewer 1 Report

This manuscript titled “Chemical characterization, sensory definition and prediction model of the cider dryness from New York State apples” (Manuscript ID: foods-2327341) explored the relationships between chemical compounds and the sensory perception of dryness of ciders and aimed to provide a sensory dryness prediction model to monitor the cider production process. The subject considered is an interesting one, and the manuscript is well organized. However, there are some issues that need to be discussed. The following are specific comments.

1.    The structure and organization of the introduction part of the manuscript is a bit cumbersome and lack of logical connection.

2.    As for the part Phenolic analysis, please describe in detail how to determine the content of tannins, flavanols, hydroxycinnamic and hydroxybenzoic acids. In addition, I wonder that how to determine that the substances detected under the same wavelength belong to the same type.

3.    Some words in the figure 1 have been covered. Please modify it.

4.    Please check the grammar of the manuscript again and pay attention to the specification of the sentence format, such as the omission of a period between two sentences in Line 17.

Author Response

Comments and Suggestions for Authors

This manuscript titled “Chemical characterization, sensory definition and prediction model of the cider dryness from New York State apples” (Manuscript ID: foods-2327341) explored the relationships between chemical compounds and the sensory perception of dryness of ciders and aimed to provide a sensory dryness prediction model to monitor the cider production process. The subject considered is an interesting one, and the manuscript is well organized. However, there are some issues that need to be discussed. The following are specific comments.

The structure and organization of the introduction part of the manuscript is a bit cumbersome and lack of logical connection.

According to the reviewer’s suggestions, we have gone through the manuscript and revised the introduction. Moreover, we have created more connection between the different paragraphs of the chapters as suggested.

As for the part Phenolic analysis, please describe in detail how to determine the content of tannins, flavanols, hydroxycinnamic and hydroxybenzoic acids. In addition, I wonder that how to determine that the substances detected under the same wavelength belong to the same type.

As suggested, we have added the requested information specifying how the calibration curve was determined and the detection wavelength of every single chemical compound determined.

Some words in the figure 1 have been covered. Please modify it.

In the figure there are many codes, so it was difficult to clearly distinguish them. However, we have tried to improve the reading by decreasing the font size and by removing as much overlaps as possible.

Please check the grammar of the manuscript again and pay attention to the specification of the sentence format, such as the omission of a period between two sentences in Line 17.

The quality of English was improved by the revision of the language. The sentences at Line 17 were corrected.

Reviewer 2 Report

The article "Chemical Characterization, Sensory Definition and Model for Predicting the Dryness of New York State Apple Cider" contributes to the development of literature for sensory analysis specialists, as well as food manufacturers offering cider products with appropriate sensory properties.

Before accepting a manuscript for publication in Foods, the following items must be corrected:

The aim is a long - very descriptive.

Methods

The methods are described accordingly. However, I would suggest writing:

The sensory tests were carried out in a sensory analysis laboratory equipped with individual cabins (temperature-controlled and combined natural/artificial light), designed in accordance with the ISO standard?

What were the test hours?

Table 2 - I would suggest writing: Reagent manufacturer and purity of reagents.

Was the scale structured or continuous?

Results

Table 4b – Statistical significance or non-significance not given

Tables (4 – 7) need to be corrected - graphically

Request

What are the strengths and limitations of this review?

Jakie sÄ… mocne strony i ograniczenia tego przeglÄ…du?

Author Response

Comments and Suggestions for Authors

The article "Chemical Characterization, Sensory Definition and Model for Predicting the Dryness of New York State Apple Cider" contributes to the development of literature for sensory analysis specialists, as well as food manufacturers offering cider products with appropriate sensory properties.

Before accepting a manuscript for publication in Foods, the following items must be corrected:

The aim is a long - very descriptive.

The research was complex and articulated in several steps, so we thought that it needed a little more descriptive and explanatory aim to help the reader.

Methods

The methods are described accordingly. However, I would suggest writing.... “The sensory tests were carried out in a sensory analysis laboratory equipped with individual cabins (temperature-controlled and combined natural/artificial light), designed in accordance with the ISO standard?”

The suggested indications were followed.

What were the test hours?

The test hours was added to the text.

Table 2 - I would suggest writing: Reagent manufacturer and purity of reagents.

As suggested, Table 2 was modified to insert the requested information.

Was the scale structured or continuous?

It was a category scale. The requested information was added to the text.

Results

Table 4b – Statistical significance or non-significance not given

As explained in the text, all the chemical analysis were carried out in three replicates and we have all the data and the relative statistical elaboration. Given the aim of the table 4 (allowing the comparison between the rating resulted by the application of the currently available scales - IRF with and without pH correction and NYCA - and the rating resulted from the trained judge’s evaluation) we have considered that it was correct to follow the usual procedure to define the dryness by those scales. This procedure provides that the score values of those scales were obtained by the ratio between two chemical values (residual sugar and malic acid average contents), and given that the table was an example of application of the overmentioned scales, standard deviation was not shown.

Concerning the dryness evaluated by the trained panel, we thought that it was more correct to consider it as a multidimensional attribute (to which all the taste and tactile descriptors could contribute), more similar to a qualitative judgement (like the typicality of the wine) than a measure of intensity. As a consequence, replicates were not considered appropriate.

Tables (4 – 7) need to be corrected - graphically

We checked the tables and correct the graphical mistakes.

Reviewer 3 Report

The paper Chemical characterization, sensory definition and prediction 2 model of the cider dryness from New York State apples studied the relationship between the production process and the customers’ sensory perception. It is an interesting study, but a few changes are required:

Spelling and grammar need improvement. Carefully check it within the entire manuscript.  

Line 41 – ``As of the last survey of January 2019`` - this part sounds old as we are in 2023. Please rephrase or change the data with more actual ones.

The passing from one paragraph to another is not smooth enough. For example, in lines 41-46 you discuss about statistical data related to cider production in the USA and cider processing units; in lines 47-53 you discuss the chemical composition of apple cider. Please improve the English language style in the entire manuscript.
Line 49 – the formulation ``when compared to 10 other commonly consumed fruits`` is incomplete.

Lines 64-66 – Be more specific here: ``High molecular weight procyanidins in ciders are known to contribute to astringency, whereas the smaller compounds contribute to bitter taste``. Clearly specify the responsible compounds.

Pay attention to text formatting – see Table 7

Author Response

Comments and Suggestions for Authors

The paper Chemical characterization, sensory definition and prediction 2 model of the cider dryness from New York State apples studied the relationship between the production process and the customers’ sensory perception. It is an interesting study, but a few changes are required:

Line 41 – ``As of the last survey of January 2019`` - this part sounds old as we are in 2023. Please rephrase or change the data with more actual ones.

As requested, we have replaced the 2019 data with more actual ones.

The passing from one paragraph to another is not smooth enough. For example, in lines 41-46 you discuss about statistical data related to cider production in the USA and cider processing units; in lines 47-53 you discuss the chemical composition of apple cider.

Thanks to the consideration of the reviewer, we have revised the introduction text and improved the connection between the different paragraphs

Please improve the English language style in the entire manuscript.  

The quality of English was improved by the revision of the language.

Line 49 – the formulation ``when compared to 10 other commonly consumed fruits`` is incomplete.

We added more information to complete the sentence.

Lines 64-66 – Be more specific here: ``High molecular weight procyanidins in ciders are known to contribute to astringency, whereas the smaller compounds contribute to bitter taste``. Clearly specify the responsible compounds.

The text was modified according to the requested information.

Pay attention to text formatting – see Table 7

We checked the tables and correct the text formatting, as suggested.

Round 2

Reviewer 1 Report

Dear Editor,

Authors have revised their manuscript, and there are no more comments now.

Best regards

Dr. Zhang Bolin